# UK Medical Cannabis Registry: A clinical outcomes analysis for insomnia

Arushika Aggarwal[1], Simon Erridge[1,2], Isaac Cowley[1], Lilia Evans[1], Madhur Varadpande[1], Evonne Clarke[2], Katy McLachlan[2], Ross Coomber[2,3], James J. Rucker[2,4,5], Mark W. Weatherall[2,6], Mikael H. Sodergren[1,2]*

1 Medical Cannabis Research Group, Imperial College London, London, United Kingdom, 2 Curaleaf Clinic, London, United Kingdom, 3 St. George's Hospital NHS Trust, London, United Kingdom, 4 Department of Psychological Medicine, Kings College London, London, United Kingdom, 5 South London & Maudsley NHS Foundation Trust, London, United Kingdom, 6 Buckinghamshire Healthcare NHS Trust, Amersham, United Kingdom

* m.sodergren@imperial.ac.uk

## Abstract

Insomnia affects approximately 10% of adults globally. Current treatments have their limitations, and there is growing evidence on the therapeutic potential of cannabis-based medicinal products for insomnia. This study aimed to assess changes in sleep-specific and general patient-reported outcome measures (PROMs) in individuals prescribed cannabis-based medicinal products for insomnia and to assess the incidence of adverse events. A case series was analysed with patients diagnosed with primary insomnia from the UK Medical Cannabis Registry (UKMCR). The primary outcome examined changes in the Single-Item Sleep Quality Scale (SQS), Generalised Anxiety Disorder-7 (GAD-7), and EuroQol-5 Dimension-5 Level (EQ-5D-5L). Changes in PROMs were assessed from baseline to 1-, 3-, 6-, 12- and 18-months. Adverse events were classified according to the CTCAE version 4.0. The inclusion criteria were met by 124 participants. SQS scores showed improvement from baseline ($2.66 \pm 2.41$) to 1- ($5.67 \pm 2.65$; $p < 0.001$), 3- ($5.41 \pm 2.69$; $p < 0.001$), 6- ($4.80 \pm 2.89$; $p < 0.001$), 12- ($4.24 \pm 3.01$; $p < 0.001$) and 18-months ($3.81 \pm 2.90$; $p < 0.001$). GAD-7 scores improved from baseline to 1-, 3-, 6-, 12- and 18-months ($p < 0.050$). There were also improvements in EQ-5D-5L dimensions of usual activities, pain/discomfort, anxiety/depression, and index values ($p < 0.001$). Eleven (8.87%) participants reported a total of 112 (90.32%) adverse events, but none were disabling or life-threatening. The study demonstrated improvements in subjective sleep quality and other captured PROMs in insomnia patients treated with cannabis-based medicinal products. Although the treatment was generally well-tolerated, randomised controlled trials are needed to confirm the effectiveness and safety of cannabis-based medicinal products.

**Data availability statement:** Data that support the findings of this study are available from the UK Medical Cannabis Registry. Restrictions apply to the availability of these data. Data specifications, and applications are available from https://ukmedicalcannabisregistry.com/contact/.

**Funding:** The authors received no specific funding for this work.

**Competing interests:** AA, IC, LE, MV have no conflicts of interest to declare. SE, EC, KM, RC, JJR, MWW, MHS are all either employees or work on as medical practitioners at Curaleaf Clinic. Through King's College London, JJR also receives grant funding from COMPASS Pathways PLC and consultancy fees from Beckley PsyTech and Clerkenwell Health. This does not alter our adherence to PLOS policies on sharing data and materials.

## Introduction

Insomnia is a prevalent sleep disorder, characterised by difficulty initiating or maintaining sleep, early-morning waking, and significant distress or impairment in daily functioning [1]. Insomnia disorder's diagnostic criteria, require these sleep difficulties to occur at least three nights per week over a period of more than three months, despite sufficient opportunities for sleep, and without attribution to another sleep-wake disorder or health condition [1]. One in three individuals are affected by insomnia symptoms [2–4], and an estimated 10% of adults meet the diagnostic criteria for insomnia disorder globally [5,6]. Chronic insomnia has been linked to negative health outcomes, including psychiatric disorders [7,8], cardiovascular diseases [9,10], and a reduction in health-related quality of life (HRQoL) [11,12]. Patients with insomnia are also more likely to be absent from work, retire on health grounds and experience a loss of productivity across multiple settings [13–15]. The increasing prevalence of insomnia is a significant public health concern [16], emphasising the need for effective interventions.

The pathophysiology of insomnia is complex and multifactorial. Cognitive and physiological hyperarousal are central features of insomnia, disrupting sleep [17]. Hyperarousal is associated with reductions in rapid eye movement (REM) and stage 3 non-REM sleep [18]. Additionally, changes in circadian rhythm can impact the ability to initiate and maintain sleep [19]. Neurochemical changes are implicated in insomnia disorder, particularly in γ-aminobutyric acid (GABA) levels, an inhibitory neurotransmitter. Increases in activity within GABAergic neurons in the preoptic area and parafacial zone in the brain are linked to promoting the initiation and consolidation of sleep [20].

The endocannabinoid system (ECS) is a complex neuromodulatory network, comprised of endogenous cannabinoids (endocannabinoids), cannabinoid receptors, and enzymes involved in their synthesis and degradation [21]. Among these receptors, cannabinoid receptor-1 (CB1R) and cannabinoid receptor-2 (CB2R) are the most prominent [21]. CB1R, primarily found in the central nervous system (CNS), has demonstrated effects on increasing both REM sleep and the stability of non-REM sleep when activated [22,23]. Furthermore, CB1R knockout mice have exhibited reduced REM and non-REM sleep, indicating a role for CB1R regulating the sleep-wake cycle [24]. Preclinical data indicates that anandamide, the endogenous ligand for CB1R, promotes stage 3 non-REM sleep, potentially through elevation of extracellular adenosine levels [25,26].

The first-line treatment for insomnia disorder is cognitive behavioural therapy for insomnia (CBT-I) [27]. However, there is limited accessibility due to a shortage of trained CBT-I providers [28] and an absence of clear referral pathways [29]. Medications which increase GABAergic activity, such as benzodiazepines and non-benzodiazepines (Z-drugs), are commonly prescribed for insomnia [30,31]. While the short-term efficacy of benzodiazepines and Z-drugs is well supported, evidence for long-term benefit is limited. Six-month RCTs of eszopiclone offer some support [32], but concerns about dependence, withdrawal symptoms and other adverse

effects remain [33–36]. Despite this, an estimated 300,000 patients in the UK have been prescribed benzodiazepines and Z-drugs for longer than the recommended duration of 4 weeks [37]. Melatonin is frequently used for the management of insomnia, due to its role in regulating the sleep-wake cycle and its effectiveness in promoting sleep through activation of melatonin 1a and 1b receptors [38,39]. However, despite melatonin demonstrating a favourable tolerability profile [40], it remains unclear whether its effects on sleep are of clinical significance [41,42]. More recently, dual orexin receptor antagonists have been introduced as a novel class of hypnotics. These agents promote sleep by inhibiting wakefulness, and early evidence suggests they may offer a favourable safety profile compared to traditional GABAergic medications [43,44]. However, many of these medications are unavailable in Europe. Daridorexant, the only orexin antagonist available in Europe, has recently been approved by the National Institute for Health and Care Excellence as a second-line treatment for insomnia disorder [45]. However, the use of this medication class remains limited by cost, availability, and safety data [46,47]. Considering the challenges in effectively treating insomnia disorder, only 27% of individuals achieve remission [48]. Therefore, there is a growing interest in exploring whether cannabis-based medicinal products and modulation of the ECS may have therapeutic potential for insomnia disorder.

The primary active pharmaceutical ingredients of cannabis-based medicinal products are cannabidiol (CBD) and $\Delta^9$-tetrahydrocannabinol (THC) [49]. Like anandamide, THC is a partial agonist of CB1R [50], whereas CBD is a negative allosteric modulator [51]. CBD has demonstrated the ability to increase anandamide concentrations, through inhibition of fatty acid binding proteins, which are responsible for transporting anandamide for enzymatic degradation [52,53]. There is therefore interest in determining if these compounds have similar sleep-promoting effects.

Increasing evidence evaluating various preparations supports the potential benefits of cannabis-based medicinal products for insomnia disorder [54–56]. A recent trial evaluating the safety and efficacy of a cannabinoid extract containing THC 20mg/mL, CBD 1mg/mL, and cannabinol 2mg/mL for 2 weeks demonstrated an improvement in sleep quality and symptoms in 23 individuals with insomnia [57]. Similar findings have been corroborated by preclinical studies [58–60]. Furthermore, a newly formulated repeat-action tablet comprising THC and cannabinol has exhibited an association with an improvement in sleep-quality in a cohort of 35 participants [61]. These findings were derived from a combination of both objective sleep measurements with a validated sleep-tracking system, and from subjective data gathered from participant completed questionnaires [61]. These novel formulations and their beneficial impact on sleep quality provide promise for the use of cannabis-based medicinal products for primary insomnia, but more studies are required to determine the optimal formulation for improved sleep. A systematic review of 34 studies found that every study documented improvements in sleep quality among a portion of participants [62]. This review considered patients receiving treatment for conditions other than insomnia and there was no subgroup analysis, again prohibiting identification of the optimal cannabis-based medicinal products for treatment of insomnia. Meanwhile, cannabis-based medicinal products are generally considered clinically safe, with minimal occurrences of severe adverse events (AEs) reported [55,57,63].

Whilst the research is promising and many individuals turn to cannabis for symptom relief from sleep disorders [64–66], there is a paucity of high-quality clinical data on the efficacy of cannabis-based medicinal products [67,68]. Although studies have suggested across various conditions that cannabis-based medicinal products are efficacious in improving sleep quality as a secondary outcome [69–72], there is insufficient evidence on the effects in individuals primarily treated for insomnia disorder [62]. Studies evaluating cannabis-based medicinal product use for insomnia disorder as a secondary condition likely introduce confounding variables, such as participant heterogeneity and variability in efficacy outcomes [56]. This is also likely attributed to methodological variability across the literature, including concentrations of different cannabinoids, routes of administration and differences in cannabis-based medicinal product formulations. However, studies suggest that a higher ratio of CBD to THC has been found to be more effective in managing insomnia symptoms [54,68].

Current studies in insomnia disorder are limited by small sample sizes, bias stemming from dependence on self-reported measures, and short follow up, and there is a critical need for further large-scale randomised controlled trials

(RCTs) [73–75]. Concerns exist regarding potential tolerance to THC and CBD [60,76,77], effects with prolonged administration, and multiple studies indicate that sleep disturbances occur during withdrawal following chronic cannabis use, with decreases in sleep efficiency, total sleep time and REM sleep [78,79]. Therefore, there is a need for longitudinal studies to assess the long-term impact of cannabis use. Due to the absence of appropriately designed RCTs with extended follow-up periods, there are currently no guidelines supporting the widespread population-based use of cannabis-based medicinal products for treating insomnia. Whilst upcoming RCTs such as the 'CANSLEEP' and 'CUPID' trials hold promise for investigation into the impact of cannabis on insomnia disorder [80,81], real-world evidence can offer valuable insights that can help guide future research and contribute to clinical practice guidelines.

## Aims

The study primarily aims to assess the changes in sleep-specific and general patient-reported outcome measures (PROMs) in individuals who are prescribed cannabis-based medicinal products for insomnia disorder, for those enrolled on the UK Medical Cannabis Registry (UKMCR). The secondary aim is to assess the incidence of AEs in these patients.

## Methods

### Study design

This study was a case series of individuals prescribed cannabis-based medicinal products for the primary indication of insomnia disorder, from the UKMCR. The Central Bristol Research Ethics Committee (reference: 22/SW/0145) granted ethical approval to the UKMCR for this study. Informed and written consent was obtained from each participant, prior to consecutive enrolment.

### Setting and participants

Founded in 2019 and under private ownership by Curaleaf Clinic, the UKMCR is one of the most extensive patient registries dedicated to prospectively collected pseudonymised data from patients prescribed cannabis-based medicinal products [82]. Data is gathered during clinical interactions and through a bespoke electronic reporting tool for patients in the United Kingdom and Crown Dependencies [82].

All cannabis-based medicinal product prescriptions were issued in accordance with UK regulations, which required individuals to have a previously confirmed diagnosis of insomnia that did not improve after two or more licensed medications [83]. Additionally, a consultant physician assessed suitability for treatment in conjunction with a multi-disciplinary team.

Participants were required to be at least 18 years old, diagnosed with primary insomnia, and initiated on cannabis-based medicinal product therapy for insomnia, to meet the inclusion criteria. Participants were excluded if they did not complete PROMs at baseline. Additionally, those who initiated cannabis-based medicinal product less than 18 months before data extraction from the UKMCR (December 13, 2023) were excluded.

### Data collection

Upon registration, patient demographics including age, gender, body mass index (BMI) and occupation were recorded. The Charlson Comorbidity Index, a widely used and validated tool [84], commonly utilised in registries to predict comorbidity [85], was used to determine a score for each patient based off their provided medical history.

Data on alcohol consumption and tobacco use was collected. Cannabis status was categorised as: 'current user,' 'ex-user,' or 'never used.' To measure lifetime cannabis consumption among current and ex-users, the gram years metric was used. This metric, utilised in previous UKMCR studies [86–88], is determined by multiplying the average daily cannabis consumption in grams by the number of years of use [89]. Patients were counselled to stop external sources of cannabis during treatment with CBMPs, hence cannabis consumption outside of legally prescribed use was not recorded.

Information regarding cannabis-based medicinal product prescription, including dosage, strain, formulation, administration route and manufacturing company, was directly documented from every prescription. Cannabis-based medicinal product dosages were calculated by multiplying the concentration (mg/g or mg/ml) with the prescribed daily dose (g/day or ml/day). In cases where prescriptions provided a range for both concentration and daily dose (e.g., concentration of 250–270 mg/g and daily dose of 0.50-1.00 g/day), the median value was used (e.g., 260 mg/g and 0.75 g/day in the given example).

## Outcome measures

The primary outcomes were the changes in PROM scores from baseline to follow-up at 1-, 3-, 6-, 12-, and 18-months. Secondary outcomes included the incidence and severity of AEs.

Employing PROMs for self-assessment is considered beneficial in evaluating sleep disorders [90]. Baseline PROMs were electronically recorded before the initial prescription for cannabis-based medicinal products was issued. These PROMs were then repeated at 1-, 3-, 6-, 12-, and 18-months, and were compared to baseline to facilitate comparison with the patient's condition before commencing cannabis-based medicinal product therapy. A description of the PROMs used, their scoring system, and their psychometric properties is contained within Table 1.

Table 1. Description, scoring and psychometric properties of each patient reported outcome measures (PROM).

| PROM | Description | Scoring | Psychometric properties |
|---|---|---|---|
| **Single-item sleep quality scale (SQS)** | The SQS is used to assess an individual's perception of their overall sleep quality. It involves a single question asking participants to rate their sleep quality over the past week [91]. | Respondents rate their sleep quality on a scale from 0-10 where 0 represents 'terrible' sleep quality and 10 represents 'excellent' sleep quality [91]. A mean change in score of 2.6 from baseline to follow-up is considered the MCID [91]. | The SQS when compared to the morning questionnaire-insomnia (MQI) shows excellent concurrent validity (Pearson correlation = -0.76) and a moderate test-retest reliability (intraclass correlation = 0.62) [91]. |
| **Generalised Anxiety Disorder-7 (GAD-7)** | The GAD-7 is used to screen for and assess the severity of generalised anxiety disorder. It consists of 7 items that inquire about symptoms experienced over the past 2 weeks, such as feeling nervous, anxious or worried [92]. | Each item is rated on a scale of 0–3, with higher scores indicating greater anxiety severity. The total score ranges from 0-21, with scores of ≥5, ≥10 and ≥15 representing mild, moderate, and severe levels of anxiety, respectively [92]. A 4-point change from baseline to follow-up is considered the MCID [93]. | The GAD-7 shows excellent internal consistency (Cronbach α = 0.92) and good test-retest reliability (intraclass correlation = 0.83) [92]. |
| **EuroQol-5 Dimension-5 Level (EQ-5D-5L)** | The EQ-5D-5L is a widely used HRQoL questionnaire consisting of 5 dimensions: mobility, self-care, usual activities, pain/discomfort and anxiety/depression [94,95]. | Each dimension is scored on a scale from 1-5, with 1 for 'no problem" to 5 for 'extreme problems/unable to' [96]. The scores can be joined to give a 5-digit number and converted to an index number where 1 is the best health state and <0 is perceived as worse than death [97]. | Test-retest reliability has shown to be good (intraclass correlation ≥0.8 and ≥0.7) [96]. Internal consistency cannot be assessed for preference-based measures like this one) [96]. |
| **Patient Global Impression of Change (PGIC)** | The PGIC is used to assess a patient's perception of their overall improvement or deterioration in health status over time. It involves a single question asking patients to rate their perception of change since the start of treatment [98]. | This is scored on a 7-point scale, where 1 represents 'no change (or condition has got worse) and 7 represents 'a great deal better' [99]. | Over a variety of conditions, test-retest reliability has demonstrated an intraclass correlation ranging from 0.53-0.85 [100]. |

The description column explains what the PROM is assessing, and the scoring column clarifies the meaning behind the ratings provided by the patients. The minimal clinically important difference (MCID) is the minimum change in score from baseline to follow-up, to be considered clinically significant. Psychometric properties refer to the reliability and validity of the clinical tool.

### Adverse events

Patient-reported AEs were either self-disclosed remotely or documented by their clinician during regular visits. AEs were classified according to the Common Terminology Criteria for Adverse Events version 4.0 [101].

### Missing data

A baseline observation carried forward (BOCF) approach was employed to address incomplete PROM data from patient dropout or incompletion of questionnaires. Presuming that discontinuation of cannabis-based medicinal product therapy would result in participants reverting to their initial PROM scores, missing data was substituted with participants' baseline PROM scores.

### Statistical analysis

Baseline demographics, alcohol and recreational drug consumption, comorbidities and AEs were assessed using descriptive statistics. Parametric data is shown as the mean ± standard deviation (SD), while nonparametric data is shown as the median [interquartile range (IQR)].

To assess changes in PROM scores across the analysis, a linear mixed effects model was utilised. Effect size was reported as $\eta^2$ values. Subsequently, significant findings linear mixed effects model underwent post-hoc pairwise comparisons, with the application of Bonferroni correction. Effect size for pairwise comparisons were reported as Cohen's d values and interpreted as small (d ≥ 0.2), medium (d ≥ 0.5), and large (d ≥ 0.8), respectively [102]. This analysis diminished the likelihood of type I errors [103,104]. PROM data was considered parametric, based on the central limit theorem [105].

A univariate logistic regression analysis was conducted to assess how collected variables relate to the probability of reporting a minimal clinically important difference (MCID) at the 18-month follow-up for both the SQS and GAD-7. A further multivariate analysis was conducted, which also considered the influence of other variables in attaining the MCID, for a more comprehensive analysis. Logistic regression was also planned to be conducted on individual prevalence of AEs. Results of the logistic regression are presented as odds ratios (ORs) and 95% confidence intervals (CIs).

Statistical analyses were conducted using the Statistical Package for the Social Sciences (SPSS; v.29.0.0.0), while GraphPad Prism (v. 9.4.1(350)) was utilised for graph creation. P < 0.050 indicated statistical significance.

## Results

On the date of data extraction (December 13, 2023), 19,763 patients were enrolled on the UKMCR. Following application of inclusion criteria, 124 were included in the present analysis (Fig 1).

### Baseline demographics

Table 2 displays the full demographic information of the study participants. Eighty-seven (70.16%) participants were male, and 37 (29.84%) participants were female. The mean age was 42.99 (± 13.43) years and the mean BMI was 27.40 (± 5.90) kg/m². Ninety-eight (79.03%) participants were employed, whilst 25 (20.16%) were unemployed.

Table 3 details the full medical history collected for study participants. The most prevalent comorbidity among participants was 'anxiety and/or depression' with 27 (21.77%) participants. The Charlson Co-morbidity Index had a median score of 0.00 [0.00-4.50]. Thirty-six (29.03%) participants were current smokers, 58 (46.77%) were ex-smokers and 30 (24.19%) had never smoked. With regards to cannabis status, 72 (58.06%) participants were current users, 26 (20.97%) were ex-users and 26 (20.97%) had never used. The median cannabis gram years for current or ex-users was 8.00 [2.00-22.00].

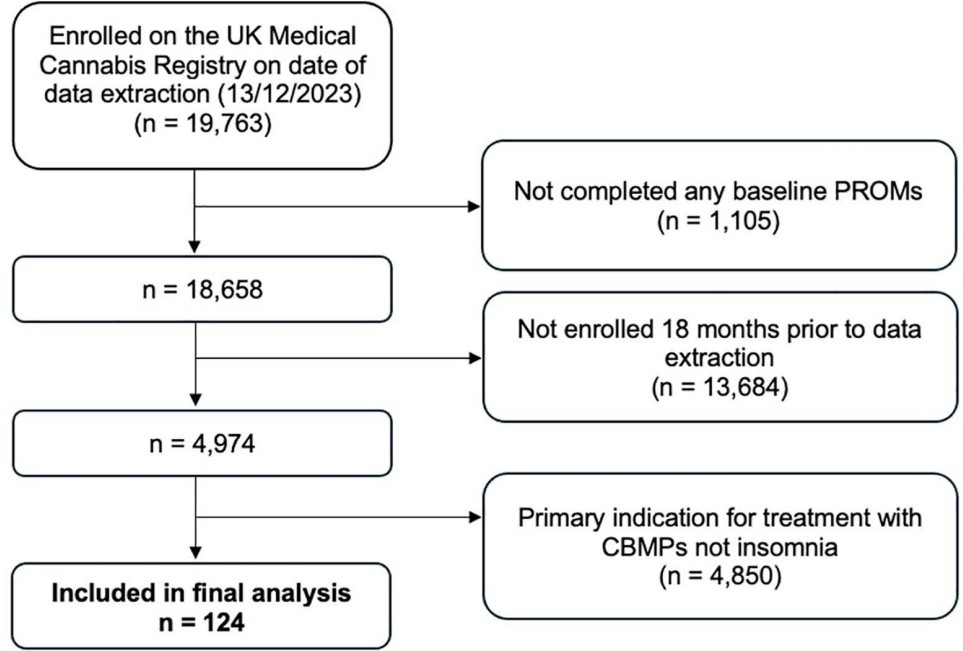

**Fig 1. Flowchart illustrating the inclusion and exclusion criteria applied to individuals enrolled in the UK Medical Cannabis Registry (UKMCR) on the date of data extraction.** The diagram depicts the number of participants excluded at each step and the reasons for exclusion, resulting in 124 patients eligible for analysis. PROMs – patient-reported outcome measures.

## Cannabis-based medicinal products

Details of cannabis-based medicinal product treatment at baseline and the maximum titrated dose were available for all participants (n = 124) (Table 4). Administration routes were also available at baseline (n = 124), follow-up months 1, 3, 6, and 12 (n = 123) and 18-months (n = 124). The median daily CBD dose at baseline was 1.00 [0.00-20.00] mg/day and increased to 10.00 [0.00-25.00] mg/day by month 3, and this was sustained until 18-month follow-up (10.00 [5.00-35.75] mg/day). For THC, the median daily dose was 20.00 [2.00-20.00] mg/day at baseline, and by 18-month follow-up, increased to 120.00 [95.00-210.38] mg/day. The most prescribed regimen at baseline (n = 51; 41.13%) and throughout every follow-up month until month 18 (n = 54; 43.55%) was dried flower only.

## Patient-reported outcome measures

A linear mixed effects model was utilised to compare PROM scores across all recorded time periods (Table 5). Single-Item Sleep Quality Score (SQS) improved from baseline (2.66±2.41) to 1- (5.67±2.65; p < 0.001; d = 1.07), 3- (5.41±2.69; p < 0.001; d = 0.96), 6- (4.80±2.89; p < 0.001; 0.76), 12- (4.24±3.01; p < 0.001; d = 0.59) and 18- (3.81±2.90; p < 0.001; d = 0.47) months (Fig 2a; S1 Appendix). Generalised Anxiety Diorder-7 (GAD-7) scores improved from baseline (9.59±6.31) to 1-month (4.99±4.91; p < 0.001; d = 0.81). The differences between baseline and follow-up months 3, 6, 12 and 18 were less pronounced over time, but still statistically significant (p < 0.050) (Fig 2b; S2 Appendix). The EuroQol-5 Dimension-5 Level (EQ-5D-5L) dimensions of pain/discomfort (p = 0.036), anxiety/depression (p = 0.001), and index values (p < 0.001) all demonstrated significance. All pairwise comparisons for these PROMs are detailed in full in S3–S5 Appendices However, no significant changes were observed in the EQ-5D-5L mobility (p = 0.966), self-care (p = 0.983), or usual

**Table 2. Baseline demographic data of the study population.**

| Baseline demographics | n (%)/ mean ± SD |
|---|---|
| Sex | |
| Male | 87 (70.16%) |
| Female | 37 (29.84%) |
| Age (years) | 42.99 ± 13.43 |
| BMI (kg/m²) | 27.40 ± 5.90 |
| Occupation | |
| Employed | 98 (79.03%) |
| Clerical support workers | 3 (2.42%) |
| Craft and related trades workers | 7 (5.65%) |
| Elementary occupations | 6 (4.84%) |
| Managers | 6 (4.84%) |
| Other occupations | 19 (15.32%) |
| Professional | 20 (16.13%) |
| Service and sales workers | 7 (5.65%) |
| Skilled agricultural, forestry and fishery workers | 4 (3.23%) |
| Technicians and associate professionals | 14 (11.29%) |
| Unspecified | 12 (9.68%) |
| Retired | 1 (0.81%) |
| Unemployed | 25 (20.16%) |

The data includes sex distribution, age, body mass index (BMI) and occupation categories, represented as counts, percentages, or mean values with standard deviation (SD). (n = 124).

activities (p = 0.208) dimensions. Patient Global Impression of Change (PGIC) scores did not show significant differences across the follow-up period (p = 0.591).

## Adverse events

A total of 112 (90.32%) adverse events were reported by 11 (8.87%) participants (Table 6). The common adverse events were fatigue (n = 11; 8.87%), insomnia (n = 11; 8.87%) and dry mouth (n = 9; 7.26%). There were no disabling or life-threatening adverse events.

## Univariate and multivariate analysis

A univariate logistic regression analysis was performed to evaluate the association between collected variables and the likelihood of reporting a minimal clinically important difference (MCID) at the 18-month follow-up point for the SQS (S6 Appendix). It was found that none of the variables increased the odds of improving SQS scores. A multivariate analysis was also conducted (S7 Appendix), which revealed that the age category of >50 years was associated with improved likelihood of attaining the MCID (OR = 49.49; 95% CI: 1.50 – 1635.25; p = 0.029).

 A univariate analysis was also conducted to evaluate the variables which may have contributed to reaching the MCID for the GAD-7 at 18-month follow-up (S8 Appendix). Analysis revealed none of the variables were associated with improved odds of reporting a clinically significant reduction in GAD-7 scores. A subsequent multivariate analysis was performed, which revealed the age category of 31–40 (OR = 13.52; 95% CI: 1.20 – 152.54; p = 0.035), and the THC dosage being above the median of 120mg/day (OR = 0.24; 95% CI: 0.06 – 0.97; p = 0.045) were associated with an increase likelihood of improved anxiety symptoms (S9 Appendix).

**Table 3. Medical history of the study population (n = 124).**

| Medical history | n (%)/ median [IQR] |
| --- | --- |
| Charlson Co-morbidity Index | 0.00 [0.00-4.50] |
| Comorbidities | |
| Myocardial Infarction | 1 (0.81%) |
| Congestive heart failure | 2 (1.61%) |
| Peripheral vascular disease | 1 (0.81%) |
| Cerebrovascular accident or transient ischemic   attack | 1 (0.81%) |
| Dementia | 0 (0%) |
| Chronic obstructive pulmonary disease | 0 (0%) |
| Connective tissue disease | 0 (0%) |
| Peptic ulcer disease | 0 (0%) |
| Liver disease | |
| Mild | 2 (1.61%) |
| Moderate to severe | 1 (0.81%) |
| Diabetes | |
| End organ damage | 1 (0.81%) |
| None or diet-controlled | 116 (93.55%) |
| Uncomplicated | 6 (4.84%) |
| Hemiplegia | 1 (0.81%) |
| Moderate to severe chronic kidney   disease | 1 (0.81%) |
| Solid tumour | |
| Metastatic | 2 (1.61%) |
| Localised | 3 (2.42%) |
| Leukaemia | 0 (0%) |
| Lymphoma | 0 (0%) |
| AIDs | 0 (0%) |
| Hypertension | 8 (6.45%) |
| Anxiety and/or depression | 27 (21.77%) |
| Arthritis | 3 (2.42%) |
| Epilepsy | 3 (2.42%) |
| Venous thromboembolism | 0 (0%) |
| Endocrine dysfunction | 4 (3.23%) |
| Smoking status | |
| Never smoked | 30 (24.19%) |
| Ex-smoker | 58 (46.77%) |
| Current smoker | 36 (29.03%) |
| Smoking pack years (current or ex-smokers) | 10.00 [2.45-20.00] |
| Weekly alcohol consumption (units) | 2.00 [0.00-10.00] |
| Cannabis status | |
| Never used | 26 (20.97%) |
| Ex-user | 26 (20.97%) |
| Current user | 72 (58.06%) |
| Cannabis grams per day (current users) | 1.00 [1.00-2.00] |
| Cannabis gram years (current or ex-users) | 8.00 [2.00-22.00] |

The data includes the Charlson Co-morbidity Index, comorbidities, smoking status, smoking pack years, weekly alcohol consumption, cannabis status, and cannabis gram years. The data are expressed as counts (%) or median values [interquartile range (IQR)].

**Table 4. Data on prescribed cannabis-based medicinal products recorded for participants (n = 124).**

| Cannabis-based medicinal product details | Baseline | 1-month | 3-months | 6-months | 12-months | 18-months |
|---|---|---|---|---|---|---|
| | | | n (%)/ median [IQR] | | | |
| **Prescription information** | | | | | | |
| CBD dosage (mg/day) | 1.00 [0.00-20.00] | 7.50 [0.00-20.00] | 10.00 [0.00-25.00] | 10.00 [3.13-25.00] | 10.00 [5.00-26.13] | 10.00 [5.00-35.75] |
| THC dosage (mg/day) | 20.00 [2.00-20.00] | 98.75 [10.00-105.00] | 100.00 [16.25-149.06] | 110.00 [80.41-195.00] | 122.50 [83.75-212.00] | 120.00 [95.00-210.38] |
| **Administration routes** | | | | | | |
| No. of patients taking oils/equivalent | 41 (33.06%) | 37* (30.08%) | 34* (27.64%) | 28* (22.76%) | 26* (21.14%) | 26 (20.97%) |
| No. of patients taking dried flower | 51 (41.13%) | 48* (39.02%) | 47* (38.21%) | 48* (39.02%) | 56* (45.53%) | 54 (43.55%) |
| No. of patients taking both | 32 (25.81%) | 38* (30.89%) | 42* (34.15%) | 47* (38.21%) | 41* (33.33%) | 44 (35.48%) |

This includes prescription information represented as median [interquartile range (IQR)], and administration routes, represented as counts (%). *One participant was removed from cannabis-based medicinal product analysis to prevent inadvertent re-identification (n = 123). CBD – cannabidiol; THC - $\Delta^9$-tetrahydrocannabinol.

**Table 5. Results of patient-reported outcome measures (PROMs) scores across different time points.**

| PROM | Baseline | 1 month | 3 months | 6 months | 12 months | 18 months | F statistic | df | p-value | η² |
|---|---|---|---|---|---|---|---|---|---|---|
| SQS | 2.66±2.41 | 5.67±2.65 | 5.41±2.69 | 4.80±2.89 | 4.24±3.01 | 3.81±2.90 | 19.96 | 5, 732 | <0.001*** | 0.12 |
| GAD-7 | 9.59±6.31 | 4.99±4.91 | 5.47±5.26 | 6.73±5.95 | 7.42±6.16 | 8.21±6.15 | 10.84 | 5, 738 | <0.001*** | 0.068 |
| EQ-5D-5L Mobility | 1.40±0.87 | 1.42±0.82 | 1.38±0.79 | 1.37±0.79 | 1.46±0.88 | 1.37±0.78 | 0.19 | 5, 732 | 0.966 | 0.001 |
| EQ-5D-5L Selfcare | 1.32±0.70 | 1.28±0.68 | 1.31±0.70 | 1.31±0.68 | 1.32±0.66 | 1.35±0.75 | 0.14 | 5, 732 | 0.983 | 0.001 |
| EQ-5D-5L Usual activities | 1.85±1.04 | 1.59±0.91 | 1.61±0.96 | 1.70±1.01 | 1.80±1.03 | 1.78±1.03 | 1.44 | 5, 732 | 0.208 | 0.01 |
| EQ-5D-5L Pain/Discomfort | 2.09±1.12 | 1.74±0.84 | 1.76±0.88 | 1.89±0.99 | 2.00±1.07 | 1.95±0.99 | 2.39 | 5, 732 | 0.036* | 0.016 |
| EQ-5D-5L Anxiety/depression | 2.63±1.24 | 2.03±0.99 | 2.15±1.14 | 2.28±1.16 | 2.36±1.21 | 2.41±1.19 | 4.06 | 5, 732 | 0.001** | 0.027 |
| EQ-5D-5L Index value | 0.65±0.29 | 0.76±0.21 | 0.74±0.24 | 0.72±0.25 | 0.68±0.27 | 0.69±0.25 | 3.19 | 5, 732 | 0.007** | 0.021 |
| PGIC | NA | 5.51±1.42 | 5.71±1.30 | 5.76±1.20 | 5.75±1.26 | 5.74±1.36 | 0.7 | 4, 570 | 0.591 | 0.005 |

Data analysed using Linear Mixed Effects Model. The PROMs include GAD-7 (Generalised Anxiety Disorder-7), SQS (Single-Item Sleep Quality Scale), EQ-5D-5L (EuroQol-5 Dimension-5 Level) dimensions (Mobility, Selfcare, Usual Activities, Pain/discomfort, Anxiety/depression, Index values) and PGIC (Patient Global Impression of Change). The data are expressed as mean ± standard deviation. df – degrees of freedom, *p < 0.050, ** p < 0.010, ***p < 0.001.

Due to a small number of only 11 participants reporting AEs, logistic regression analysis could not be feasibly conducted to ascertain factors that lead to increased odds of AEs.

## Discussion

This case series focussed on a cohort of patients treated with cannabis-based medicinal products for a diagnosis of insomnia. The study demonstrated improvements in subjective sleep quality, anxiety and HRQoL PROMs. Across multiple metrics including SQS, GAD-7 and EQ-5D-5L dimensions (usual activities, pain/discomfort, anxiety/depression, and index values), improvements were observed from baseline to 1-, 3-, 6-, 12- and 18-month follow-ups. Fewer than 1

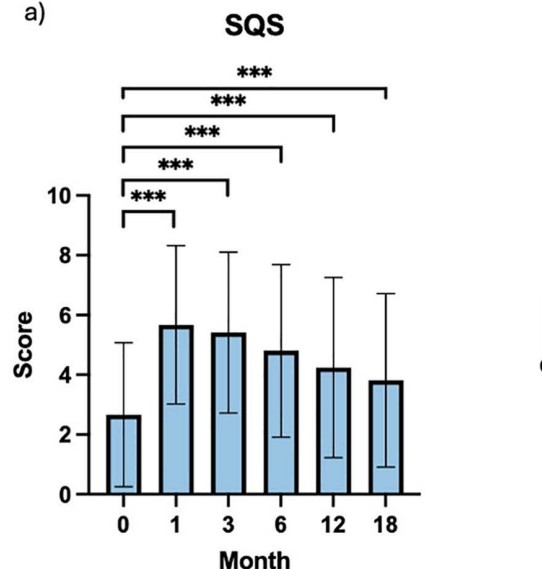
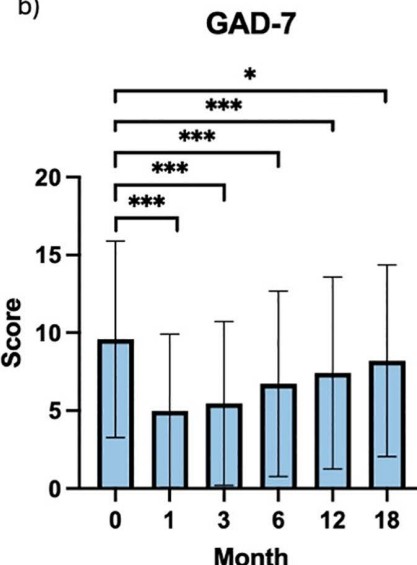

**Fig 2. Bar charts showing a) Single-Item Sleep Quality Scale (SQS) and b) Generalised Anxiety Disorder-7 (GAD-7) scores at baseline (0 months) and at subsequent intervals (1-, 3-, 6-, 12- and 18-months).** Error bars indicate one standard deviation above and below mean. Statistical analysis by repeated measures analysis of variance (ANOVA) and post-hoc pairwise comparisons with Bonferroni correction. *$p < 0.050$, **$p < 0.010$, ***$p < 0.001$. (n = 124).

in 10 participants reported an adverse event, most of which were categorised as mild or moderate, with no disabling or life-threatening events. These results must be interpreted within the acknowledged limitations of study design, however.

In a prior UKMCR study employing a similar methodology to assess patients treated with cannabis-based medicinal products from primary insomnia, an improvement in SQS scores from baseline up to 6-months was observed [55], mirroring findings of the present study. However, the previous study found a consistent size of change between baseline and follow-up scores across 6-months. In contrast, the present study revealed that while the magnitude of improvement and effect size was most prominent at 1-month, there was a decline in the magnitude of the change between baseline and follow-up scores, with this trend persisting up to month 18. This may be due to the larger sample size in the present study, as well as the longer follow-up period, showing more variability in responses over time. Moreover, within the shorter follow-up duration of the previous study, it is plausible that the anticipation or expectation of cannabis effects could have contributed to an enhanced SQS score [106]. One study has indicated the presence of an exaggerated placebo effect with cannabis, where patients report subjective experiences which decrease following repeated exposure [107]. This phenomenon might explain a more uniform change between baseline and follow-up scores observed in the initial months. Additionally, the diminishing magnitude of improvement observed between baseline and subsequent follow-up months in our study may be attributed to tolerance. Research indicates that tolerance can develop to the sedative effects of THC [108], potentially accounting for the reduced improvement in subjective sleep quality over time. Notably, this apparent reduction in treatment effect over time coincided with increasing doses of both CBD and THC, suggesting that participants may have developed tolerance to the therapeutic effects of cannabis-based medicinal products. This dose-response shift further supports the possibility of pharmacological tolerance contributing to the diminishing improvements observed in SQS and GAD-7 scores. Similar results to the present study were found in a 3-month study by Shannon *et al.*, where CBD administration demonstrated no lasting improvements in sleep scores, and the percentage of patients reporting an improvement in sleep decreased over the study period [109]. However, an important consideration is the use of the BOCF

**Table 6. Frequency and severity of adverse events reported during the study. Adverse events are categorised as mild, moderate, or severe with corresponding counts provided for each event category. The total percentage of adverse events for each severity level is presented. (n = 124).**

| Adverse event | Mild | Moderate | Severe | Total (%) |
|---|---|---|---|---|
| Abdominal pain | 1 | 0 | 0 | 1 (0.81%) |
| Agitation | 0 | 0 | 1 | 1 (0.81%) |
| Amnesia | 1 | 1 | 1 | 3 (2.42%) |
| Anorexia | 0 | 1 | 0 | 1 (0.81%) |
| Ataxia | 1 | 1 | 0 | 2 (1.61%) |
| Blurred vision | 1 | 0 | 0 | 1 (0.81%) |
| Cognitive disturbance | 1 | 1 | 0 | 2 (1.61%) |
| Concentration impairment | 3 | 1 | 1 | 5 (4.03%) |
| Confusion | 1 | 1 | 0 | 2 (1.61%) |
| Constipation | 2 | 0 | 0 | 2 (1.61%) |
| Cough | 1 | 1 | 0 | 2 (1.61%) |
| Delirium | 1 | 1 | 1 | 3 (2.42%) |
| Diarrhoea | 0 | 0 | 1 | 1 (0.81%) |
| Dizziness | 4 | 4 | 0 | 8 (6.45%) |
| Dry mouth | 7 | 2 | 0 | 9 (7.26%) |
| Dysgeusia | 0 | 2 | 0 | 2 (1.61%) |
| Dyspepsia | 2 | 3 | 0 | 5 (4.03%) |
| Fall | 0 | 1 | 0 | 1 (0.81%) |
| Fatigue | 4 | 7 | 0 | 11 (8.87%) |
| Fever | 0 | 1 | 0 | 1 (0.81%) |
| Headache | 5 | 0 | 0 | 5 (4.03%) |
| Insomnia | 0 | 6 | 5 | 11 (8.87%) |
| Lethargy | 6 | 2 | 0 | 8 (6.45%) |
| Nausea | 4 | 0 | 0 | 4 (3.23%) |
| Palpitations | 0 | 2 | 0 | 2 (1.61%) |
| Pharyngitis | 0 | 2 | 0 | 2 (1.61%) |
| Somnolence | 0 | 7 | 1 | 8 (6.45%) |
| Tremor | 2 | 0 | 0 | 2 (1.61%) |
| Vertigo | 5 | 1 | 0 | 6 (4.84%) |
| Weight loss | 1 | 0 | 0 | 1 (0.81%) |
| **Total (%)** | **53 (42.74%)** | **48 (38.71%)** | **11 (8.87%)** | **112 (90.32%)** |

approach for missing data. As this approach replaces missing data with baseline PROM scores, this may cause an artificial maintenance of PROM scores as the follow-up time increases, causing the magnitude of improvement in scores and effect size to decrease over time. Additionally, the BOCF approach disregards changes seen in patients who discontinued due to factors not associated with the treatment [110].

Whilst the present study demonstrated improvements in sleep quality following cannabis-based medicinal product treatment, a recent RCT assessing the effect of cannabis-based medicinal products for primary insomnia revealed that the treatment yielded comparable results to placebo across most sleep-related outcomes [75]. The variation in response could be attributed to differences in cannabis-based medicinal product formulations. While the current study employed cannabis-based medicinal products containing both CBD and THC, the study by Narayan *et al.* solely looked at the response to 150mg CBD [75]. Meanwhile, a comparison of combined THC and CBD resulted in greater perceived

improvement of insomnia symptoms, compared to using THC or CBD individually in a separate study [111]. Furthermore, whilst higher doses of CBD and THC have been shown to enhance total sleep duration, lower doses have been shown to promote wakefulness [111,112]. The effect of this may be potentially seen in the current study, where the doses of CBD and THC taken by patients increased from baseline to 18-month follow-up, suggesting the higher doses were more beneficial for insomnia symptoms. Additionally, whilst THC has been shown to decrease sleep latency [56], there are concerns of tolerance with prolonged use [76]. This impact on sleep latency offers a potential reason for the improvement in SQS score, and similar results have been seen in other RCTs [57,113]. However, these RCTs also incorporate cannabinol as part of their cannabis-based medicinal product formulation. Cannabinol may have been present in preparations within the present study, but at low levels and uncaptured by the UK Medical Cannabis Registry. Moreover, other studies use other PROMs to assess sleep quality other than the SQS. The considerable heterogeneity across the literature highlights the necessity for further RCTs to investigate the varying effects of different cannabis-based medicinal product formulations on sleep outcomes in individuals with insomnia.

At follow-up periods, participants reported a reduction in anxiety symptoms, evidenced by improvements in GAD-7 scores. This mirrors findings from previous UKMCR studies [55,114,115]. Preclinical research has provided insights into the dual effects of cannabinoids, exhibiting both anxiolytic and anxiogenic properties, and highlighted the involvement of the endocannabinoid system in regulating emotional behaviour and anxiety-related responses [116], despite the paucity of RCTs investigating cannabis-based medicinal product efficacy for generalised anxiety disorders. Insomnia and anxiety are believed to share a bidirectional relationship [117], as hyperarousal often results in poor sleep, and this has been demonstrated to cause dysregulation of the hypothalamic-pituitary-adrenal (HPA) axis [118]. This can result in heightened levels of anxiety [118,119]. Even in the present study, the most prevalent comorbidity was 'anxiety and/or depression'. However, due to limited sample size in the current analysis, it was not feasible to conduct a subgroup analysis to determine whether the presence of anxiety influences the response to cannabis-based medicinal products prescribed for insomnia. Nevertheless, it appears that cannabis-based medicinal product treatment holds promise in positively impacting both conditions that often manifest as secondary to each other.

Despite several studies indicating that insomnia is associated with lower scores in HRQoL domains [12,120,121], there is insufficient evidence to determine whether cannabis-based medicinal products improve these scores in individuals with insomnia. The present study utilises several dimensions of the EQ-5D-5L questionnaire to determine HRQoL, which has been shown to be a valid and reliable tool for various conditions [122,123]. An observational study assessing the impact of cannabis-based medicinal products on HRQoL with a 3-month follow-up found that overall, they were associated with clinically significant improvements in EQ-5D-5L index scores, consistent with findings in the present study [124]. However, the current study also showed that whilst the EQ-5D-5L dimensions of usual activities, pain/discomfort and anxiety/depression showed greatest improvement from baseline to 1-month, this magnitude of improvement decreased over the following months up to month 18. This suggests that HRQoL may eventually return to baseline over a longer period time, indicating a potential decline in cannabis-based medicinal product effectiveness, or a response shift in the patient [125]. The lack of significant difference in EQ-5D-5L mobility and self-care could be accounted for by the relatively young age of the participants, with a mean age of 42.99 years, as they typically experience fewer physical limitations. Therefore, any potential effects of cannabis-based medicinal products on these domains may have been less pronounced.

The univariate logistic regression analysis showed no significance for the SQS and GAD-7. This is likely due to the small sample size, and it has been suggested that a minimum sample size of 500 is required to derive the appropriate statistics [126]. Despite this, multivariate logistic regression analysis was conducted to account for intricate relationships amongst the variables. This approach ensured that the effects of each variable were independently assessed without being obscured by the influence of others, as might occur in univariate analysis. The multivariate analysis revealed that the age category of >50 was associated with improved odds of attaining the MCID for the SQS. From this, it could be hypothesised that older individuals may possess a sleep architecture that exhibits heightened responsiveness to

the effects of cannabinoids. Whilst there is insufficient evidence on this, it has been suggested that biological changes associated with ageing may increase cannabinoid distribution and decrease elimination, thus altering sleep outcomes [127]. Additional multivariate analysis revealed that the THC dose being above the median of 120mg/day increased the likelihood of improved anxiety symptoms, and similar results were found in a study by Stack *et al.* where THC dominant formulations demonstrated a significant improvement in anxiety symptoms [128]. However, there are mixed results in the literature, with indications that THC has a dose-dependent effect on anxiety, with anxiolytic effects at lower doses, but anxiogenic effects at higher doses [129]. Additionally, a previous UKMCR study revealed non-significance regarding the association between higher doses of THC and the likelihood of improving anxiety symptoms [115]. Overall, these conflicting findings suggest that further analysis is required to determine patient and medication specific factors associated with positive outcomes.

From a total of 112 (90.32%) AEs reported by 11 (8.87%) participants, 53 (42.74%) were mild, 48 (38.71%) were moderate and 11 (8.87%) were severe. The most common AEs were fatigue, insomnia, and dry mouth, consistent with previous studies [55,130]. Furthermore, an RCT evaluating the safety of a cannabis-based medicinal product formulation for insomnia over 2 weeks reported no serious AEs [57]. Almost all AEs reported, which were categorised as mild, had resolved either overnight or upon waking (97.5%) [57]. Generally, short-term cannabis use has demonstrated safety, and a systematic review conducted in 2008 identified that out of a total of 4779 reported adverse events, 4615 (96.6%) of these were not serious [131]. However, the risks of AEs linked to prolonged use remain inadequately understood in the literature, and higher quality trials investigating long-term exposure are needed to establish safety of cannabis-based medicinal product use. Furthermore, reliance on self-reports might have led to treatment-related effects being observed as AEs. For instance, commonly reported AEs in the present study such as insomnia, lethargy, and fatigue may be indicative of insomnia symptoms.

## Limitations

As a case series, this hinders the assessment of a causal association between cannabis-based medicinal product therapy and improvements in HRQoL and sleep-specific outcomes. Furthermore, the absence of blinding or a placebo control group diminishes internal validity and, considering the subjective nature of PROMs, may exacerbate reporting bias. The societal perceptions and media portrayal of cannabis as having positive effects may contribute to greater placebo effects, particularly among previous users who anticipate its effects [106,127].

The use of PROMs also introduces limitations. The utilisation of the SQS to evaluate sleep quality only records symptoms experienced within the past week, potentially failing to capture the longer term reduced sleep quality typically linked to insomnia. Alternatively, the insomnia severity index evaluates symptoms over the past 2 weeks, whilst the Pittsburgh sleep quality index assesses symptoms over 1-month and may be more suitable measures for evaluating sleep quality. The GAD-7 to assess the severity of generalised anxiety disorder also only captures symptoms experienced over the past 2 weeks and does not necessarily reflect the longitudinal symptoms experienced by patients. Another measure, such as the generalised anxiety disorder questionnaire-IV, assesses symptom history over the past six months. It has revealed excellent sensitivity and specificity [132], and may provide a more accurate representation of anxiety severity in insomnia. Furthermore, PROM questionnaires are subject to recall bias, as they are reliant on recollections of past events by study participants, where there are likely to be differences in accuracy. The use of PROMs which only require recall over the past 1–2 weeks, such as the SQS and GAD-7, may therefore reduce recall bias. In future studies, it would be beneficial to use a combination of both PROM scores and actigraphy or formal sleep study data [133]. This would help capture both the subjective patient experience as well as objective data on sleep and wakefulness to build a comprehensive understanding of the effects of cannabis-based medicinal products on sleep quality.

Analysing data from private medical cannabis clinics may have introduced selection bias as it is likely participants came from higher socioeconomic backgrounds. However, several studies have suggested that it is those from lower socioeconomic backgrounds who have a higher likelihood of experiencing insomnia and insomnia-related health consequences [134–136]. Furthermore, those who have previously used cannabis recreationally are more likely to use it medicinally [137]. This is evident in the present study's higher percentage of individuals who are current or ex-users of cannabis (79.03%) compared to the reported 7.1% of adults in Great Britain who had used cannabis between 2020–2021 [138]. Therefore, the findings of the present study have limited generalisability to the wider population.

As the size of the insomnia cohort within the UKMCR during this analysis was limited, this reduced the power of the study. There was also insufficient data to conduct subgroup analyses to determine whether specific patient characteristics affect cannabis-based medicinal product response in addition to the multivariate analysis. As the dataset expands, future UKMCR analyses should explore the impact of various factors, including how patient comorbidities, strains of cannabis, and formulation of cannabis-based medicinal products may influence treatment responses for primary insomnia. Unfortunately, the dataset does not capture information on the sub-type of insomnia diagnosed. In individuals with secondary insomnia to a condition that may also be addressable by cannabis-based medicinal products this may bias the results to a positive finding, as the improvement in sleep may be secondary to changes in the underlying condition. In addition there may be variation in outcomes between Furthermore, whilst this study holds the distinction of having the longest follow-up period at 18-months among those investigating cannabis-based medicinal products for insomnia, further research with even longer follow-up periods is required to establish the long-term impact of cannabis-based medicinal product treatment, including optimal dosage, safety, and potential tolerance.

## Conclusion

This case series study investigated the outcomes of insomnia patients prescribed cannabis-based medicinal products over an 18-month period. The findings indicate a promising association between cannabis-based medicinal product treatment and improvements in sleep-specific outcomes and general HRQoL measures. However, it is crucial to acknowledge that these findings must be interpreted with consideration of the limitations in the study's design, and there is a need for high quality RCTs to assess the long-term efficacy and safety of cannabis-based medicinal product for primary insomnia. Whilst the study demonstrates good tolerability and improvement in PROMs within 18-months, the findings also indicate that this improvement may not be sustained over a longer period, and tolerance could develop. These findings can be used to inform future RCTs.

## Supporting information

**S1 Appendix: Post-hoc comparison of sleep quality scale (SQS) scores for each follow-up period, after significance was determined with the linear mixed effects model.** The values represent the mean difference ± standard error. *p < 0.050, **p < 0.010, ***p < 0.001. d - Cohen's d.
(DOCX)

**S2 Appendix: Post-hoc comparison of generalised anxiety disorder-7 (GAD-7) scores for each follow-up period, after significance was determined with the linear mixed effects model.** The values represent the mean difference ± standard error. *p < 0.050, **p < 0.010, ***p < 0.001. d - Cohen's d.
(DOCX)

**S3 Appendix: Post-hoc comparison of European quality-of-life-5 dimension-5 level (EQ-5D-5L) pain/discomfort scores for each follow-up period, after significance was determined with the linear mixed effects model.** The values represent the mean difference ± standard error. *p < 0.050, **p < 0.010, ***p < 0.001. d - Cohen's d.
(DOCX)

**S4 Appendix: Post-hoc comparison of European quality-of-life-5 dimension-5 level (EQ-5D-5L) anxiety/depression scores for each follow-up period, after significance was determined with a linear mixed effects model.** The values represent the mean difference ± standard error. *p < 0.050, **p < 0.010, ***p < 0.001. d - Cohen's d.
(DOCX)

**S5 Appendix: Post-hoc comparison of European quality-of-life-5 dimension-5 level (EQ-5D-5L) index value scores for each follow-up period, after significance was determined with the linear mixed effects model.** The values represent the mean difference ± standard error. *p < 0.050, **p < 0.010, ***p < 0.001. d - Cohen's d.
(DOCX)

**S6 Appendix: Table illustrating the odds ratio (95% confidence interval (CI)), representing the impact of individual variables on participants' attainment of the minimally clinically important difference (MCID) on the sleep quality scale (SQS) at 18-month follow up.** A univariate logistic regression model was utilised to conduct statistical analysis. n = 123. CBD – cannabidiol; CBMP – cannabis-based medicinal product; THC - $\Delta^9$-tetrahydrocannabinol.
(DOCX)

**S7 Appendix: Table illustrating the odds ratio (95% confidence interval (CI)), representing the impact of variables on participants' attainment of the minimally clinically important difference (MCID) on the sleep quality scale (SQS) at 18-month follow up.** A multivariate logistic regression model was utilised to conduct statistical analysis. *p < 0.050. n = 117. CBD – cannabidiol; CBMP – cannabis-based medicinal product; THC - $\Delta^9$-tetrahydrocannabinol.
(DOCX)

**S8 Appendix: Table illustrating the odds ratio (95% confidence interval (CI)), representing the impact of individual variables on participants' attainment of the minimally clinically important difference (MCID) on generalised anxiety disorder-7 (GAD-7) at 18-month follow up.** A univariate logistic regression model was utilised to conduct statistical analysis. n = 124. CBD – cannabidiol; CBMP – cannabis-based medicinal product; THC - $\Delta^9$-tetrahydrocannabinol.
(DOCX)

**S9 Appendix: Table illustrating the odds ratio (95% confidence interval (CI)), representing the impact of variables on participants' attainment of the minimally clinically important difference (MCID) on generalised anxiety disorder-7 (GAD-7) at 18-month follow up.** A multivariate logistic regression model was utilised to conduct statistical analysis. *p < 0.050. n = 118. CBD – cannabidiol; CBMP – cannabis-based medicinal product; THC - $\Delta^9$-tetrahydrocannabinol.
(DOCX)

## Acknowledgments

**Disclosure**: Patient consent statement. All study participants gave formal, informed, and written consent, preceding their consecutive enrollment into the database.

## Author contributions

**Conceptualization:** Arushika Aggarwal, Simon Erridge, Evonne Clarke, Katy McLachlan, Ross Coomber, James J Rucker, Mark W Weatherall, Mikael H Sodergren.

**Data curation:** Arushika Aggarwal, Simon Erridge, Isaac Cowley, Lilia Evans, Madhur Varadpande, Evonne Clarke, Katy McLachlan, Ross Coomber, James J Rucker, Mark W Weatherall, Mikael H Sodergren.

**Formal analysis:** Arushika Aggarwal, Simon Erridge, Mikael H Sodergren.

**Investigation:** Arushika Aggarwal, Simon Erridge, Isaac Cowley, Lilia Evans, Madhur Varadpande, Mikael H Sodergren.

**Methodology:** Arushika Aggarwal, Simon Erridge, Mikael H Sodergren.

**Project administration:** Arushika Aggarwal, Simon Erridge, Evonne Clarke, Katy McLachlan, Ross Coomber, James J Rucker, Mark W Weatherall, Mikael H Sodergren.

**Supervision:** Simon Erridge, Mikael H Sodergren.

**Visualization:** Arushika Aggarwal, Simon Erridge, Mikael H Sodergren.

**Writing – original draft:** Arushika Aggarwal, Simon Erridge, Isaac Cowley, Lilia Evans, Madhur Varadpande, Mikael H Sodergren.

**Writing – review & editing:** Arushika Aggarwal, Simon Erridge, Isaac Cowley, Lilia Evans, Madhur Varadpande, Evonne Clarke, Katy McLachlan, Ross Coomber, James J Rucker, Mark W Weatherall, Mikael H Sodergren.

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
