## [Decision Letter · Decision Letter 0]

PMEN-D-25-00007

UK Medical Cannabis Registry: A Clinical Outcomes Analysis for Insomnia

PLOS Mental Health

Dear Dr. Sodergren,

Thank you for submitting your manuscript to PLOS Mental Health and we apologise for the delay in sending a decision to you. After careful consideration of the reviewer reports, which we have now obtained, we feel that your article has merit but does not yet fully meet PLOS Mental Health’s publication criteria as it currently stands. Therefore, we invite you to submit a revised version of the manuscript that addresses the points raised during the review process.

Please ensure that all of the reviewer comments are addressed in your revision and please feel free to reach out to me directly if you have any questions.

We look forward to receiving your revised manuscript.

Kind regards,

Karli Montague-Cardoso

Executive Editor

PLOS Mental Health

Additional Editor Comments (if provided):

Reviewers' comments:

Reviewer's Responses to Questions

**Comments to the Author**

1. Does this manuscript meet PLOS Mental Health’s publication criteria?

Reviewer #1: Yes

Reviewer #2: Yes

2. Has the statistical analysis been performed appropriately and rigorously?

Reviewer #1: Yes

Reviewer #2: No

3. Have the authors made all data underlying the findings in their manuscript fully available (please refer to the Data Availability Statement at the start of the manuscript PDF file)?

Reviewer #1: Yes

Reviewer #2: No

4. Is the manuscript presented in an intelligible fashion and written in standard English?

Reviewer #1: Yes

Reviewer #2: Yes

Reviewer #1: The manuscript "UK Medical Cannabis Registry: A Clinical Outcomes Analysis for Insomnia" addresses a common problem and a common treatment that is poorly understood and remains a public health concern. As a result, the paper is timely. The use of a public registry that contains systematically collected data on patient diagnoses and treatment is a strength. The methods, statistics, results and discussion are well-written and I have very few quibbles with any of it.

Here are my remaining concerns which are mostly minor.

1. Introduction: to say that there is no data supporting the long term efficacy and safety of Z-drugs is incorrect. Please cite the 6-month RCTs on eszopiclone

2. Introduction: The dual orexin receptor antagonists were completely overlooked as a treatment option and should be described as a safer alternative to z drugs and maybe safer than any form of cannabis

3. Introduction: There is a mention of a review of CBD/THC in 34 included studies, showing "benefit in a subset of patients". Please clarify the characteristic of the subset which benefited and who did not benefit

4. Discussion: The authors correctly describe the initial improvement then followed by a noticeable regression of benefit in the SQS and GAD7 scores, suggesting that this might represent tolerance. This impression is further underscored by the fact that CBD and THC doses are advancing at a time that SQS and GAD7 scores are regressing. The authors should explicitly link these trends as further evidence of possible tolerance of effect

Reviewer #2: The use of a national registry provides important, patient-level data that supplements the relatively limited literature that examines the efficacy of medical cannabis for insomnia. The use of several key outcomes strengthens the work, although a single-use sleep item is a weakness. There are some significant limitations in the chosen analytical plan and lack of detail that undermine the study conclusions. Re-analysis using more sophisticated analytical framework is recommended, along with correct reporting of analytical model parameters and effect size estimations.

Abstract:

- Please note that p = 0.05 is not significant per the traditional threshold. Correct this throughout and update abstract

Introduction:

- The acronym CBMP is not commonly used and detracts from the work. Please reconsider using this unfamiliar acronym.

- The authors cite that a systematic review of 34 studies found a uniform improvement in insomnia symptoms “in a sub-set of participants”. This is a bit misleading as only 2 of the 34 studies focused on patients with insomnia, and others were just poor sleep etc. Please revise this to more accurately reflect the results of the review and the samples used.

Methods:

- While I understand why no subgroup analyses were performed on patient characteristics (e.g. anxiety), what about cannabis use characteristics (e.g. administration route, THC dose etc.). Authors state that this information was available but appears to be missing from analyses

- Was information available on the sub-type of insomnia diagnosed? If not, can the authors speculate on how this might affect results and provide further discussion of this.

Analysis:

- I recommend that the authors reconduct their analyses using a more sophisticated Linear Mixed Effect Model with appropriate covariance structure, stratified by administration route. This will provide more sensitive tool to assess both time and participant-level factors for the cannabis treatments, accounting for repeated assessment timepoints.

- I am surprised that analyses did not control for baseline characteristics, such as cannabis dose and/or administration. Please include these when performing re-analysis.

- Can the authors provide more justification for not stratifying the results by treatment modality, such as administration route? Given that previous work has shown at least some effect differences for those who consume an oil-based product vs dried flower for sleep benefits, and as this information was seemingly available to the authors, it is surprising that this was not considered.

- Provide a measure of effect size for all the reported analyses (and for any updated analyses that are performed in subsequent revisions

Results:

- Was any data collected on recreational cannabis usage?

- Please provide the appropriate F-statistic, measure of effect size and df for the PROM analysis.

**Do you want your identity to be public for this peer review?** For information about this choice, including consent withdrawal, please see our Privacy Policy

Reviewer #1: No

Reviewer #2: No

---

## [Decision Letter · Decision Letter 1]

PMEN-D-25-00007R1

UK Medical Cannabis Registry: A Clinical Outcomes Analysis for Insomnia

PLOS Mental Health

Dear Dr. Sodergren,

Thank you for submitting your manuscript to PLOS Mental Health. After careful consideration of the revisions and reviewer feedback, we feel that the paper is greatly improved. We would like to invite you to make one final round of minor revisions in light of the remaining questions from the reviewer, which you can find below. Thank you for your patience and understanding

We look forward to receiving your revised manuscript.

Kind regards,

Karli Montague-Cardoso

Executive Editor

PLOS Mental Health

Journal Requirements:

Additional Editor Comments (if provided):

Reviewers' comments:

Reviewer's Responses to Questions

**Comments to the Author**

Reviewer #2: (No Response)

publication criteria?

Reviewer #2: Yes

3. Has the statistical analysis been performed appropriately and rigorously?

Reviewer #2: No

4. Have the authors made all data underlying the findings in their manuscript fully available (please refer to the Data Availability Statement at the start of the manuscript PDF file)?

Reviewer #2: Yes

5. Is the manuscript presented in an intelligible fashion and written in standard English?

Reviewer #2: Yes

Reviewer #2: Thank you for your careful review of the statistical approach. The application of LMM has strengthened the work, however key details from this approach are still missing. Can the authors please provide:

i.) how model selection was guided

ii.) what covariance structure was used and why

iii.) the random factor

iv.) the fixed factor(s)

v.) whether separate models were build per outcome

vi). whether both main effect and interactions were explored, and in what order

**Do you want your identity to be public for this peer review?** For information about this choice, including consent withdrawal, please see our Privacy Policy

Reviewer #2: No

---

## [Editor Report · Decision Letter 2]

UK Medical Cannabis Registry: A Clinical Outcomes Analysis for Insomnia

PMEN-D-25-00007R2

Dear Sodergren,

We are pleased to inform you that your manuscript 'UK Medical Cannabis Registry: A Clinical Outcomes Analysis for Insomnia' has been provisionally accepted for publication in PLOS Mental Health.

Best regards,

Karli Montague-Cardoso

Staff Editor

PLOS Mental Health